# Asymmetrical reliability of the Alda score favours a dichotomous representation of lithium responsiveness

Abraham Nunes[1,2], Thomas Trappenberg[2], Martin Alda[1¤]*, The international Consortium on Lithium Genetics (ConLiGen)[¶]

**1** Department of Psychiatry, Dalhousie University, Halifax, Nova Scotia, Canada, **2** Faculty of Computer Science, Dalhousie University, Halifax, Nova Scotia, Canada

¤ Current address: QEII Health Sciences Centre, Halifax, Nova Scotia, Canada
¶ Membership list can be found in the Acknowledgments section.
* malda@dal.ca

**Data Availability Statement:** All relevant data are within the manuscript and its Supporting Information files.

## Abstract

The Alda score is commonly used to quantify lithium responsiveness in bipolar disorder. Most often, this score is dichotomized into "responder" and "non-responder" categories, respectively. This practice is often criticized as inappropriate, since continuous variables are thought to invariably be "more informative" than their dichotomizations. We therefore investigated the degree of informativeness across raw and dichotomized versions of the Alda score, using data from a published study of the scale's inter-rater reliability (n = 59 raters of 12 standardized vignettes each). After learning a generative model for the relationship between observed and ground truth scores (the latter defined by a consensus rating of the 12 vignettes), we show that the dichotomized scale is more robust to inter-rater disagreement than the raw 0-10 scale. Further theoretical analysis shows that when a measure's reliability is stronger at one extreme of the continuum—a scenario which has received little-to-no statistical attention, but which likely occurs for the Alda score $\geq 7$—dichotomization of a continuous variable may be more informative concerning its ground truth value, particularly in the presence of noise. Our study suggests that research employing the Alda score of lithium responsiveness should continue using the dichotomous definition, particularly when data are sampled across multiple raters.

## Introduction

The Alda score is a validated index of lithium responsiveness commonly used in bipolar disorder (BD) research [1]. This scale has two components. The first is the "A" subscale that provides an ordinal score (from 0 to 10, inclusive) of the overall "response" in a therapeutic trial of lithium. The second component is the "B" subscale that attempts to qualify the degree to which any improvement was causally related to lithium. The total Alda score is computed based on these two subscale scores, and takes integer values between 0 and 10. Many studies that employ the Alda score as a target variable dichotomize it, such that individuals with scores $\geq 7$ are classified as "responders," and those with scores $< 7$ are "non-responders."

**Funding:** Genome Canada (MA, AN; https://www.
genomecanada.ca), Dalhousie Department of
Psychiatry Research Fund (MA, AN; https://
medicine.dal.ca/departments/department-sites/
psychiatry.html), Canadian Institutes of Health
Research #64410 (MA; http://www.cihr-irsc.gc.ca),
Natural Science and Engineering Research Council
of Canada (TT; https://www.nserc-crsng.gc.ca),
Nova Scotia Health Research Foundation Scotia
Scholars Graduate Scholarship (AN; https://nshrf.
ca), Killam Postgraduate Scholarship (AN; http://
www.killamlaureates.ca) and Dalhousie Medical
Research Foundation and the Lindsay family (AN
and MA). The funders had no role in study design,
data collection and analysis, decision to publish, or
preparation of the manuscript.

**Competing interests:** The authors have declared
that no competing interests exist.

A common criticism that arises from this practice is that continuous variables should not be discretized by virtue of "information loss." Indeed, discretizing continuous variables is widely viewed as an inappropriate practice [2–12]. However, the practice remains common across many areas of research, including our group's work on lithium responsiveness in BD [13]. The primary justification for using the dichotomized Alda score as the lithium responsiveness definition has been based on the inter-rater reliability study by Manchia et al. [1], who showed that a cut-off of 7 had strong inter-rater agreement (weighted kappa 0.66). Furthermore, using mixture modeling, they also found that the empirical distribution of Alda scores supports the discretized definition. Therefore, there exist competing arguments regarding the appropriateness of dichotomizing lithium response. Resolving this dispute is critical, since the operational definition of lithium responsiveness is a concept upon which a large body of research will depend.

Although the Manchia et al. [1] analysis provides some justification for using a dichotomous lithium response definition, it does not dispel the argument of discretization-induced information loss entirely. However, there is some intuitive reason to believe that discretization is, at least pragmatically, the best approach to defining lithium response using the Alda score. First, the Alda score remains inherently subjective to some degree and is not based on precise biological measurements; an individual whose "true" Alda score is 6, for example, could have observed scores that vary widely across raters. Second, it is possible that responders may be more reliably identified than non-responders. For example, unambiguously "excellent" lithium response is a phenomenon that undoubtedly exists in naturalistic settings [14, 15], and for which the space of possible Alda scores is substantially smaller than for non-responders; that is, an Alda score of 8 can be obtained in far fewer ways than an Alda score of 5. As such, we hypothesize that agreement on the Alda score is higher at the upper end of the score range, and that this asymmetric agreement is a scenario in which dichotomization of the score is more informative than the raw measure. To evaluate this, we present both empirical re-analysis of the ConLiGen study by Manchia et al. [1] and analyses of simulated data with varying levels of asymmetrical inter-rater reliability.

## Materials and methods

### Data

Detailed description of data and collection procedures is found in Manchia et al. [1]. Samples included in our analysis are detailed in Table 1, including the number of raters included across sites, and the average ratings obtained at each of those sites across the 12 assessment vignettes. As a gold standard, we used ratings that were assigned to each case vignette using a consensus process at the Halifax site (scores are noted in the first row of Table 1). The lithium responsiveness inter-rater reliability data are available in S1 File (total Alda score), and S2 File (Alda A-score).

### Empirical analysis of the Alda score

In this analysis, we seek to evaluate whether discretization of the Alda score under the existing inter-rater reliability values preserves *more* mutual information (MI) between the observed and ground truth labels than does the raw scale representation. To accomplish this, we first develop a probabilistic formulation of raters' score assignments based on a multinomial-Dirichlet model, which we describe below. Since the Dirichlet distribution is the conjugate prior for the multinomial distribution, the posterior distribution over ratings (and ultimately the MI with respect to the "ground truth" Alda score) can be expressed as a closed-form function of the prior uncertainty, which increases the precision and efficiency of our experiments.

**Table 1. Number of raters and mean scores across sites.** The total number of raters ($n_r$) was 59.

| Site | $n_r$ | Case Vignette | | | | | | | | | | | |
|---|---|---|---|---|---|---|---|---|---|---|---|---|---|
| | | 1 | 2 | 3 | 4 | 5 | 6 | 7 | 8 | 9 | 10 | 11 | 12 |
| Gold standard | | 8 | 9 | 6 | 7 | 9 | 3 | 5 | 9 | 3 | 9 | 5 | 1 |
| Halifax | 9 | 8.4 | 8.6 | 6.6 | 6.9 | 9.2 | 3 | 3.9 | 8.8 | 3.1 | 9.1 | 4.7 | 1.2 |
| NIMH | 4 | 7.8 | 8.2 | 6.2 | 7 | 8.8 | 3.2 | 4 | 8.5 | 2.2 | 8.5 | 3.2 | 1.8 |
| Poznan | 2 | 9 | 8.5 | 6.5 | 5.5 | 9 | 4 | 7.5 | 9 | 5 | 8 | 4.5 | 4.5 |
| Dresden | 2 | 8.5 | 7.5 | 6 | 5 | 8.5 | 1.5 | 6 | 9 | 3.5 | 8.5 | 4 | 1.5 |
| Japan | 4 | 8 | 8.2 | 4.8 | 6.5 | 8.5 | 2 | 3 | 8.5 | 1 | 8.2 | 4.5 | 1.5 |
| Wuerzburg | 2 | 7.5 | 7.5 | 4 | 6.5 | 8 | 1.5 | 3 | 9 | 0 | 7 | 3 | 0.5 |
| Cagliari | 3 | 7.7 | 9 | 4.3 | 7 | 5.7 | 4 | 1.3 | 9 | 0.7 | 7.3 | 4 | 2 |
| San Diego | 2 | 7.5 | 8.5 | 7.5 | 7 | 9 | 5 | 7.5 | 8.5 | 3.5 | 8.5 | 6 | 3.5 |
| Boston | 2 | 8.5 | 8.5 | 6 | 7 | 9 | 3 | 3.5 | 8.5 | 1.5 | 9 | 4 | 1 |
| Gottingen | 2 | 9.5 | 9 | 4 | 6 | 9 | 1 | 1 | 9 | 1.5 | 9 | 4 | 3 |
| Berlin | 1 | 7 | 9 | 4 | 6 | 9 | 2 | 3 | 8 | 0 | 7 | 0 | 2 |
| Taipeh | 1 | 8 | 8 | 5 | 8 | 9 | 5 | 6 | 9 | 4 | 9 | 8 | 1 |
| Prague | 1 | 7 | 9 | 4 | 8 | 9 | 3 | 6 | 9 | 3 | 9 | 6 | 1 |
| Johns Hopkins | 7 | 8 | 8.7 | 5.3 | 5.9 | 8.3 | 2.7 | 2.4 | 9.1 | 2 | 8.3 | 4.4 | 1.1 |
| Mayo | 6 | 8 | 8.2 | 6 | 8 | 9 | 4.2 | 3 | 9 | 4.2 | 8.8 | 3.7 | 0.3 |
| Brasil | 3 | 8 | 8.3 | 5.3 | 6.3 | 8.7 | 2 | 4 | 9 | 4.3 | 8 | 4.7 | 0.7 |
| Medellin | 4 | 7.5 | 9 | 5.5 | 6.5 | 5 | 2.5 | 4 | 7.2 | 4.8 | 8.8 | 1.2 | 2 |
| Geneve | 3 | 7.7 | 8.7 | 6.7 | 5.3 | 9.7 | 5 | 6 | 8.7 | 1.3 | 9 | 3.7 | 0.3 |

Let $n_i^{(k)} \in \mathbb{N}_+$ denote the number of raters who assigned an Alda score $i \in \mathcal{A}$, with $\mathcal{A} = \{0, \ 1, \ ..., \ 10\}$ to an individual whose gold standard Alda score is $k \in \mathcal{A}$. The vector of rating counts for the gold standard score $k$ is is $\mathbf{n}^{(k)} = (n_i^{(k)})_{i \in \mathcal{A}}$. The probability of $\mathbf{n}^{(k)}$ is multinomial with parameter vector $\boldsymbol{\theta}^{(k)} = (\theta_i^{(k)})_{i \in \mathcal{A}}$, which is itself Dirichlet distributed $\boldsymbol{\theta}^{(k)} \sim \mathrm{Dir}(\boldsymbol{\theta}|\boldsymbol{\alpha})$, where $\boldsymbol{\alpha}$ is a pseudocount denoting the prior expectation of the number of ratings received for each score $i \in \mathcal{A}$. In the present analysis, we assume that $\boldsymbol{\alpha}$ is equal across all scores in $\mathcal{A}$, and thus we denote it simply as a scalar $\boldsymbol{\alpha} = \alpha$; this has the effect of increasing the uncertainty of $\boldsymbol{\theta}^{(k)}$ (i.e. the ratings become more "noisy").

The posterior of $\boldsymbol{\theta}^{(k)}$ given $\mathbf{n}^{(k)}$ and $\alpha$ is Dirichlet with parameters $\boldsymbol{\alpha}' = \left\{\alpha + n_i^{(k)} - 1\right\}_{i=0}^{10}$, and its *maximum a posteriori* (MAP) estimate is

$$\hat{\boldsymbol{\theta}}_\alpha(\mathbf{n}^{(k)}) = \left\{ \frac{\alpha + n_i^{(k)} - 1}{\sum_{j=0}^{10} \alpha + n_j^{(k)} - 1} \right\}_{i=0}^{10}, \tag{1}$$

which can be viewed as the conditional distribution over scores $\mathcal{A}$ for any given rater when the gold standard is $k$. In cases where no assessment vignette had a gold standard rating of $k$, we assumed that

$$\mathbf{n}^{(k)} = \begin{cases} \frac{1}{2}\left(\mathbf{n}^{(k-1)} + \mathbf{n}^{(k+1)}\right) & 0 < k < 10 \\ \mathbf{n}^{(k+1)} & k = 0 \\ \mathbf{n}^{(k-1)} & k = 10 \end{cases} \tag{2}$$

The dichotomized Alda scores are defined as $T = \{\delta[i \geq \tau] : \forall i \in \mathcal{A}\}$, where $\tau$ is the dichotomization threshold (set at $\tau = 7$ for the Alda score), and where $\delta[\cdot]$ is an indicator function that evaluates to 1 if the argument is true, and 0 otherwise. Given threshold $\tau$ (Responders $\geq \tau$ and Non-responders $< \tau$), the dichotomous counts are represented as follows

$$
\begin{aligned}
c_0^{(0)} &= \sum_{k=0}^{\tau-1} \sum_{i=0}^{\tau-1} n_i^{(k)} \quad && \text{Observed} \; < \tau, \; \text{Gold Standard} \; < \tau \\
c_0^{(1)} &= \sum_{k=\tau}^{10} \sum_{i=0}^{\tau-1} n_i^{(k)} \quad && \text{Observed} \; < \tau, \; \text{Gold Standard} \; \geq \tau \\
c_1^{(0)} &= \sum_{k=0}^{\tau-1} \sum_{i=\tau}^{10} n_i^{(k)} \quad && \text{Observed} \; \geq \tau, \; \text{Gold Standard} \; < \tau \\
c_1^{(1)} &= \sum_{k=\tau}^{10} \sum_{i=\tau}^{10} n_i^{(k)} \quad && \text{Observed} \; \geq \tau, \; \text{Gold Standard} \; \geq \tau
\end{aligned}
\tag{3}
$$

with $\mathbf{c}^{(k)} \sim \text{Multinomial}(\phi_k)$, and $\phi_k \sim \text{Dir}(\phi|\xi)$, where $\xi$ is a pseudocount for the number of dichotomized ratings assigned to each of non-responders and responders. We can thus estimate the conditional distribution over observed dichotomized response ratings as

$$
\hat{\phi}_\xi(\mathbf{c}^{(k)}) = \left\{ \frac{\xi + c_0^{(k)} - 1}{2\xi - 2 + c_0^{(k)} + c_1^{(k)}}, \frac{\xi + c_1^{(k)} - 1}{2\xi - 2 + c_0^{(k)} + c_1^{(k)}} \right\}
\tag{4}
$$

**Mutual information of raw and dichotomized Alda score representations.** Mutual information is a general measure of dependence that expresses the degree to which uncertainty about one variable is reduced by observation of another. Whereas the correlation coefficient depends on the existence of a linear association, MI can detect nonlinear relationships between variables by comparing their joint probability against the product of their marginal distributions.

Let

$$
x_o \sim p(x_o|x_*) = \text{Categorical}\left( \hat{\boldsymbol{\theta}}_\alpha(\mathbf{n}^{(x_*)}) \right)
\tag{5}
$$

denote a given observed *raw* Alda score assigned to a case with ground truth score of $x_* \in \mathcal{A}$. Given uniform priors on the true classes, the joint distribution is

$$
p(x_o, x_*) = p(x_o|x_*)p(x_*) = \left\{ \frac{1}{11} \hat{\boldsymbol{\theta}}_\alpha(\mathbf{n}^{(x_*=k)}) \right\}_{k=0,1,\ldots,10}.
\tag{6}
$$

For the binarized classes, we have a prior of $p(y_* = 1) = \frac{4}{11}$, and the joint distribution is thus

$$
p(y_o, y_*) = p(y_o|y_*)p(y_*) = \left\{ p(y_* = k) \; \hat{\phi}_\xi(\mathbf{c}^{(y_*=k)}) \right\} k \in \{0, 1\}.
\tag{7}
$$

The MI for these distributions can be computed as functions of the prior pseudocounts $\alpha$ and $\xi$:

$$
I_\alpha[x_o||x_*] = \sum_{x_o} \sum_{x_*} p(x_o, x_*) \log \frac{p(x_o, \, x_*)}{p(x_o)p(x_*)}
\tag{8}
$$

$$
I_\xi[y_o||y_*] = \sum_{y_o} \sum_{y_*} p(y_o, y_*) \log \frac{p(y_o, y_*)}{p(y_o)p(y_*)}
\tag{9}
$$

for the raw and dichotomized Alda scores, respectively. We can express the MI of the raw and dichotomized Alda score distributions both in terms of $\alpha$, such that both distributions have an equivalent total "concentration:" $\xi = \alpha$ when $\xi = 11\alpha/2$. This is equivalent to saying that our prior assumption about the uncertainty of the raw and dichotomized distributions assumes the same number of a priori ratings.

Our primary hypothesis—that the dichotomized Alda score is more informative with greater observation uncertainty—is evaluated by determining whether $I_\xi[y_o||y_*]$ exceeds $I_\alpha[x_o||x_*]$ as we increase the *a priori* observation noise ($\alpha$ and $\xi$).

## Theoretical modeling of dichotomization under asymmetrical reliability

The previous experiment regarding dichotomization of the raw Alda score did not fully capture the effect of dichotomization of a continuous variable, since the raw Alda score is still discrete (albeit with a larger domain of support). Thus, we sought to investigate whether dichotomization of a truly continuous, though asymmetrically reliable, variable would show a similar pattern of preserving MI and statistical power under higher levels of observation noise and agreement asymmetry.

**Synthetic datasets.** The simplest synthetic dataset generated was merely a sample of regularly spaced points across the [0, 10] interval in both the x and y directions. This dataset was merely used to conduct a "sanity check" that our methods for computing MI correctly identified a value of 0. This was necessary since data with uniform random noise over the same interval will only yield MI of 0 in the limit of large sample sizes.

The main synthetic dataset accepted "ground truth" values $x \in [0, 10]$ and yielded "observed" values $y \in [0, 10]$ based on the following formula for the $i^{\text{th}}$ sample:

$$y_i = \omega \ f(x_i) \ + \ (1 - \omega) \ \text{Uniform}(0, \ 10), \tag{10}$$

where $0 \leq \omega \leq 1$ is a parameter governing the degree to which observed values are coupled to the ground truth based on $f(x_i)$ (data are entirely uniform random noise when $\omega = 0$, and come entirely from $f(x_i)$ when $\omega = 1$). The function $f(x_i)$ governing the agreement between ground truth and observed is essentially a 1:1 correspondence between $x$ and $y$ to which we add noise along the diagonal based on a uniform random variate $\tilde{U}(-\sigma, \ \sigma)$ with width $\sigma$.

We simulated two forms of diagonal spread. The first is constant across all values $x \in [0, 10]$, which we call the *symmetrical* case, and which is represented by a parameter $\beta = 1$. The other is an *asymmetrical* case (represented as $\beta = 0$), in which the agreement between $x$ and $y$ is not constant across the [0, 10] range. Overall, the function $f(x_i)$ is defined as

$$f(x_i) \ = \beta \ R_{(0, \ 10)}\left(x_i + \frac{\tilde{U}(-\sigma, \ \sigma)}{1 + e^{-0.75 \ x_i + 5}}\right) \ + \ (1 - \beta) \ R_{(0, \ 10)}\left(x_i + \tilde{U}(-\sigma, \ \sigma)\right), \tag{11}$$

where $R_{(l, \ u)}(\cdot)$ is a function to ensure that all points remain within the [l, u] interval in both axes. In the asymmetrical case, $R_{(l, \ u)}(\cdot)$ reflects points at the [0, 10] bounds. In the symmetrical case, the data are all simply rescaled to lie in the [0, 10] interval.

Demonstration of the simulated synthetic data are shown in Fig 1. Every synthetically generated dataset included 750 samples, and for notational simplicity, we denote the $k^{\text{th}}$ synthetic dataset (given parameters $\beta$, $\omega$, $\sigma$) as $D_{\beta,\omega,\sigma}^{(k)} = \left(x_j^{(k)}, \ y_j^{(k)}\right)_{j=1,2,\ldots,750}$.

**Computation of mutual information for continuous and discrete distributions.** Mutual information was computed for both continuous and dichotomized probability distributions on the data. Mutual information for the continuous distribution was computed by first performing Gaussian kernel density estimation (using Scott's method for bandwidth

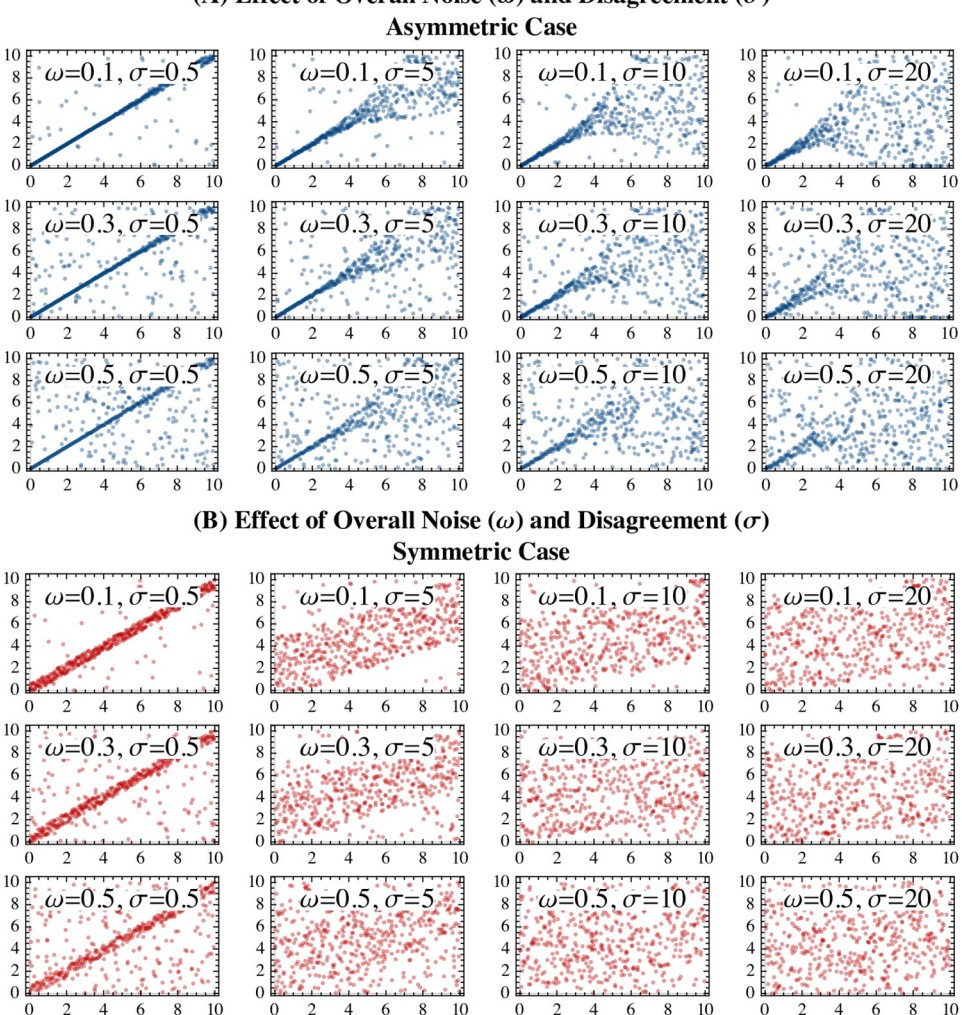

**Fig 1. Demonstration of the synthetic agreement data across differences in the parameter ranges and presence of asymmetry.** The x-axes all represent the ground truth value of the variable, and the y-axes represent the "observed" values. Data are depicted based on different values of a uniform noise parameter ($0 \leq \omega \leq 1$) that governs what proportion of the data is merely uniform noise over the interval [0, 10], and a disagreement parameter ($\sigma \geq 0$), which governs the variance around the diagonal line. Panel A (upper three rows, shown in blue) depicts the synthetic data in which there was asymmetrical levels of agreement across the score domain. Panel B (lower three rows, shown in red) depict synthetic data in which there was symmetrical agreement over the score domain.

selection) on the simulated dataset, and then approximating the following integral using Markov chain Monte-Carlo sampling:

$$I_{\text{KDE}}[y||x] = \int \int p(x,\ y) \log \frac{p(x,\ y)}{p(x)\ p(y)}\ \mathrm{d}x\ \mathrm{d}y \tag{12}$$

Conversely, discrete MI was computed by first creating a 2-dimensional histogram by binning data based on a dichotomization threshold $\tau$. Data that lie below the dichotomization threshold are denoted 1, and those that lie above the threshold are represented as 0. Based on

this joint distribution, the dichotomized MI is

$$I_\tau[y||x] = \sum_x \sum_y p(x,y) \log \frac{p(x,y)}{p(x)p(y)}. \tag{13}$$

Note that continuous MI will remain constant across $\tau$.

**Statistical power of classical tests of association.** Association between the observed ($y$) and ground truth ($x$) data can be measured using Pearson's correlation coefficient ($\rho$) when data are left as continuous, or using Fisher's exact test when data are dichotomized. The statistical power of the hypothesis that $\rho \neq 0$ given dataset $D_{\beta,\omega,\sigma}^{(k)}$ with $N^{(k)}$ observations and two-tailed statistical significance threshold $\alpha$—which here is not the same $\alpha$ used as a Dirichlet concentration in *Empirical Evaluation of the Alda Score of Lithium Response*—can be easily shown to equal

$$\text{power}_\rho(D_{\beta,\omega,\sigma}^{(k)}; \ \alpha = 0.05) = \Phi\left(|\zeta(\rho)| - \Phi^{-1}\left(1 - \frac{\alpha}{2}\right)\right), \tag{14}$$

where $\Phi(\cdot)$ and $\Phi^{-1}(\cdot)$ are the cumulative distribution function and quantile functions for a standard normal distribution, and $\zeta(\cdot)$ is Fisher's Z-transformation

$$\zeta(\rho) = \frac{1}{2} \log \frac{1+\rho}{1-\rho}. \tag{15}$$

Under a dichotomization of $D_{\beta,\omega,\sigma}^{(k)}$ with threshold $\tau$ association between the ground truth and observed data can be evaluated using a (two-tailed) Fisher's exact test, whose alternative hypothesis is that the odds ratio ($\eta$) of the dichotomized data does *not* equal 1. The null-hypothesis has a Fisher's noncentral hypergeometric distribution,

$$\Lambda_o = \text{FisherHypergeometricDistribution}\left(N_{\delta[y<\tau]}^{(k)}, \ N_{\delta[x<\tau]}^{(k)}, \ N^{(k)}, \ \eta = 1\right) \tag{16}$$

where $N^{(k)}$ is the total number of observations in sample $k$, and $N_{\delta[x<\tau]}^{(k)}$ and $N_{\delta[y<\tau]}^{(k)}$ are the number of ground truth and observed data, respectively, that fall below the dichotomization threshold $\tau$. Under the alternative hypothesis, this distribution has an odds ratio parameter estimated from the data:

$$\Lambda_a = \text{FisherHypergeometricDistribution}\left(N_{\delta[y<\tau]}^{(k)}, \ N_{\delta[x<\tau]}^{(k)}, \ N^{(k)}, \ \hat{\eta}\right). \tag{17}$$

The statistical power of Fisher's exact test under this setup and a two-tailed significance threshold of $\alpha$ is

$$\text{fp}\left(D_{\beta,\omega,\sigma}^{(k)}, \ \tau; \alpha\right) = \delta[\hat{\eta} \geq 1] \ S_{\Lambda_a}\left(S_{\Lambda_o}^{-1}\left(1 - \frac{\alpha}{2}\right)\right) + \delta[\hat{\eta} < 1]\left(1 - S_{\Lambda_a}\left(S_{\Lambda_o}^{-1}\left(\frac{\alpha}{2}\right)\right)\right), \tag{18}$$

where $S_{\Lambda_a}(\cdot)$ and $S_{\Lambda_o}^{-1}(\cdot)$ are the survival functions of the alternative hypothesis and the inverse survival function of the null hypothesis, respectively.

**Evaluation of mutual information.** The central aspect of this analysis is comparison of the dichotomized and continuous MI across values of the dichotomization threshold $\tau$, global noise $\omega$, asymmetry parameter $\beta$, and diagonal spread $\sigma$. Under all cases, we expect that increases in the global noise parameter $\omega$ will reduce the MI. We also expect that with symmetrical reliability (i.e. $\beta = 0$), the dichotomized MI will be lower than the continuous MI across all thresholds. However, as the degree of asymmetry in the reliability increases, we expect the dichotomized MI to exceed the continuous MI (i.e. as $\sigma$ increases when $\beta = 1$). Finally, as a

sanity check, we expect that both continuous and dichotomized MI will be approximately 0 when applied to a grid of points regularly spaced over the [0, 10] interval.

**Evaluation of effects on statistical power of classical tests of association.** Statistical power of the Pearson correlation coefficient and Fisher's exact test were computed across symmetrical ($\beta = 0$) and asymmetrical ($\beta = 1$) conditions of the synthetic dataset described above. Owing to the greater computational efficiency of these calculations (compared to the MI), the diagonal spread parameter was varied more densely ($\sigma \in \{1, 2, . . ., 20\}$). The power of Fisher's exact test was evaluated at two dichotomization thresholds: a median split at $\tau = 5$ and a "tail split" at $\tau = 3$. We evaluated three global noise settings ($\omega \in \{0.3, 0.5, 0.7\}$). At each experimental setting, we computed the aforementioned power levels for 100 independent synthetic datasets. Results are presented using the mean and 95% confidence intervals of the power estimates over the 100 runs under each condition. We expect that the Fisher's exact test under a "tail split" dichotomization (not a median split) will yield greater statistical power in the presence of asymmetrical reliability, greater diagonal spread, and higher global noise. However, under the symmetrically reliable condition, we expect that the statistical power will be greater for the continuous test of association.

## Materials

Mutual information experiments were conducted in Mathematica v. 12.0.0 (Wolfram Research, Inc.; Champaign, IL). Experiments evaluating the statistical power under classical tests of continuous and dichotomous association were conducted in the Python programming language. Code for analyses are also provided in S3 File (Alda score analyses), S4 File (theoretical analysis of MI under asymmetrical reliability), S5 File (theoretical evaluation of classical associative tests). The Mathematica notebooks are also available in PDF form in S6 and S7 Files.

## Results

### Empirical evaluation of the Alda score of lithium response

Histograms of the observed Alda scores for each of the gold standard vignette values are depicted in Fig 2. Resulting joint distributions of the gold standard vs. observed Alda scores (in both the raw or dichotomized representations) are shown in Fig 3 (Panels A-C) across varying levels of observation noise. Fig 3D plots the MI for the raw and dichotomized Alda scores across increasing levels of the observation noise parameter $\alpha$ (recalling that $\xi = 11\alpha/2$). Beyond an observation noise of approximately $\alpha > 3.52$, one can see that the dichotomized lithium response definition retains greater MI between the true and observed labels, compared to the raw representation.

### Discrete vs. continuous mutual information in asymmetrically reliable data

Fig 4 shows the results of the experiment on synthetic data. Under agreement levels that are constant across the $(x, y)$ domains, one can observe that MI of dichotomized representations of the variables are generally lower than their continuous counterparts. However, under asymmetrical reliability (i.e. where agreement between $x$ and $y$ decreases as $x$ increases), we see that MI is higher for the dichotomized, rather than the continuous, representations. In particular, as the level of agreement asymmetry increased (i.e. for higher values of $\sigma$), the best dichotomization thresholds decreased.

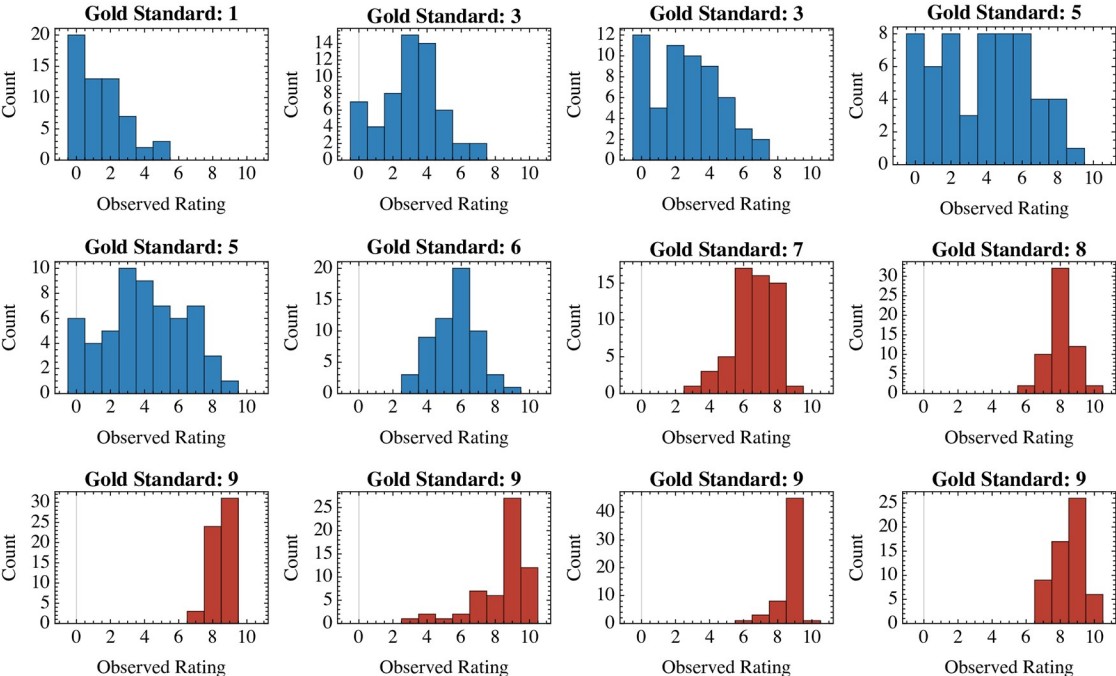

**Fig 2. Histograms of ratings for each value of the ground truth Alda score available in the first wave dataset from Manchia et al. [1].** Each histogram represents the distribution of ratings ($n_r$ = 59) for a single one of twelve assessment vignettes. The gold standard ("ground truth") Alda score, obtained by the Halifax consensus sample, is depicted as the title for each histogram. Plots in blue are those for vignettes with gold standard Alda scores less than 7, which would be classified as "non-responders" under the dichotomized setting. Vignettes with gold standard Alda scores ≥ 7 are shown in red, and represent the dichotomized group of lithium responders.

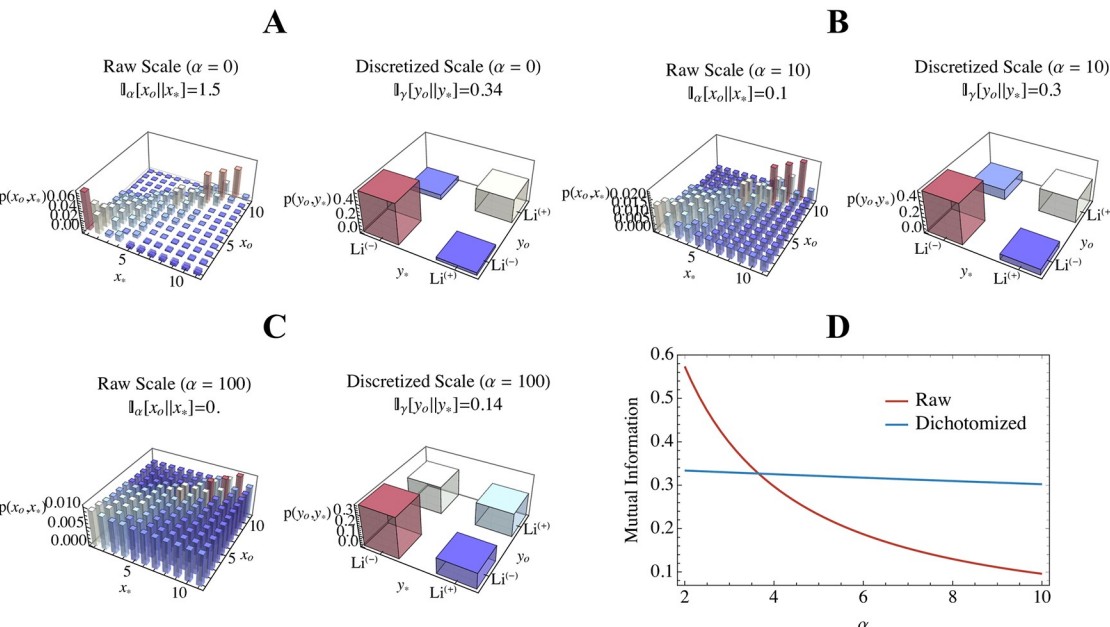

**Fig 3. Mutual information between gold standard and observed Alda scores in relation to the observation noise ($\alpha$) and whether the scale is in its raw or dichotomized form (lithium responder [Li(+)] is Alda score ≥ 7; non-responder [Li(-)] is Alda score < 7).** Panels A-C show the inferred joint distributions of the observed ($x_o$ for raw, $y_o$ for discrete) and gold standard ($x_*$ for raw, $y_*$ for discrete) values at different levels of observation noise ($\alpha \in \{0, 10, 100\}$). Panel D plots the mutual information for the raw (red) and discrete (blue) settings of the Alda score across increasing values of $\alpha$. Recall that here we set $\xi = 11/2$.

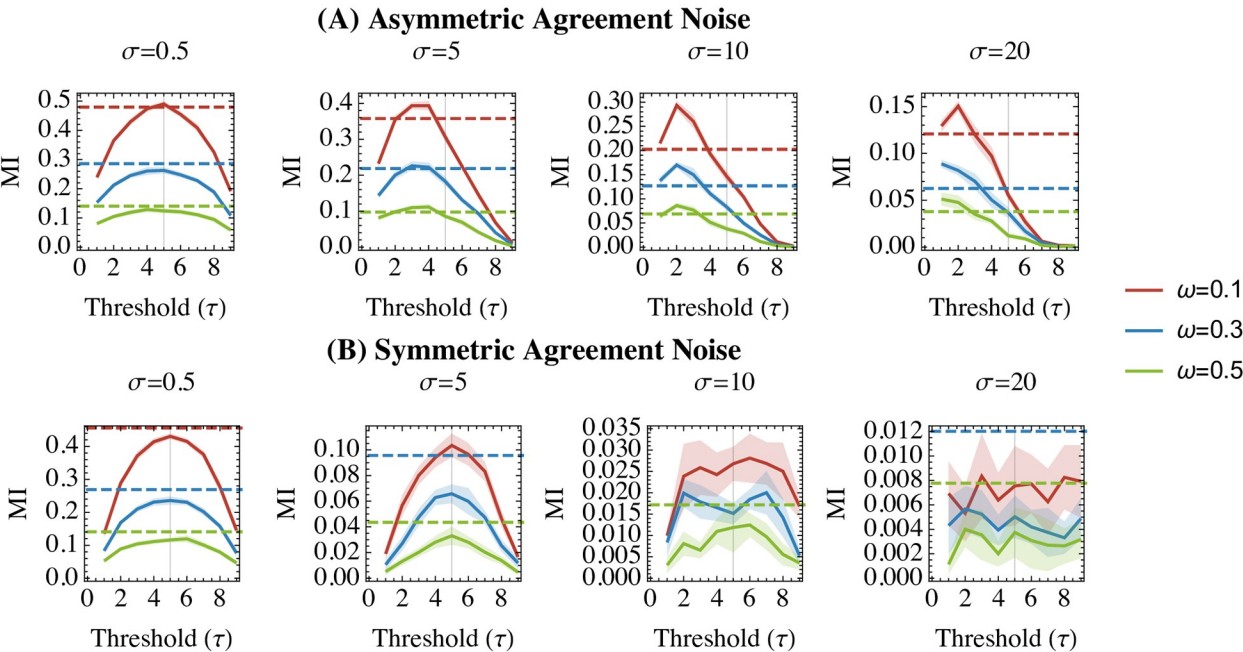

**Fig 4. Mutual information (MI) for dichotomized (solid lines) and continuous (dashed lines) distributions on synthetic data with asymmetrical (upper row, Panel A) and symmetrical (lower row, Panel B) properties with respect to agreement.** X-axes represent the dichotomization thresholds at which we recalculate the dichotomized MI. Mutual information is depicted on the y-axes. Plot titles indicate the different diagonal spread ($\sigma$) parameters used to synthesize the synthetic datasets. Solid lines (for dichotomized MI) are surrounded by ribbons depicting the 95% confidence intervals over 10 runs at each combination of parameters ($\tau, \omega, \beta, \sigma$).

## Statistical power of classical associative tests

Fig 5 plots the statistical power of null-hypothesis tests using continuous and dichotomized representations of the synthetic dataset. As expected, under conditions of symmetrical reliability, the continuous test of association (Pearson correlation) retains greater statistical power as the degree of diagonal spread increases, although this difference lessens at very high levels of diagonal spread or overall (uniform) noise. However, under conditions of asymmetrical reliability, dichotomizing data according to a "tail split" (here a threshold of $\tau = 3$) preserves greater statistical power than either a median split ($\tau = 5$) or continuous representation; this relationship was present even at high levels of diagonal spread and overall uniform noise.

## Discussion

The present study makes two important contributions. First, using a sample of 59 ratings obtained using standardized vignettes compared to a consensus-defined gold standard [1], we showed that the dichotomized Alda score has a higher MI between the observed and gold standard ratings than does the raw scale (which ranges from 0-10). Those data suggested that the Alda score's reliability is asymmetrical, with greater inter-rater agreement at the upper extreme. Secondly, therefore, using synthetic experiments we showed that asymmetrical inter-rater reliability in a score's range is the likely cause of this relationship. Our results do not argue that lithium response is itself a categorical natural phenomenon. Rather, using the dichotomous definition as a target variable in supervised learning problems likely confers greater robustness to noise in the observed ratings.

Some have argued that the existence of categorical structure in one's data [9], or evidence of improved reliability under a dichotomized structure [16], are potentially justifiable rationales

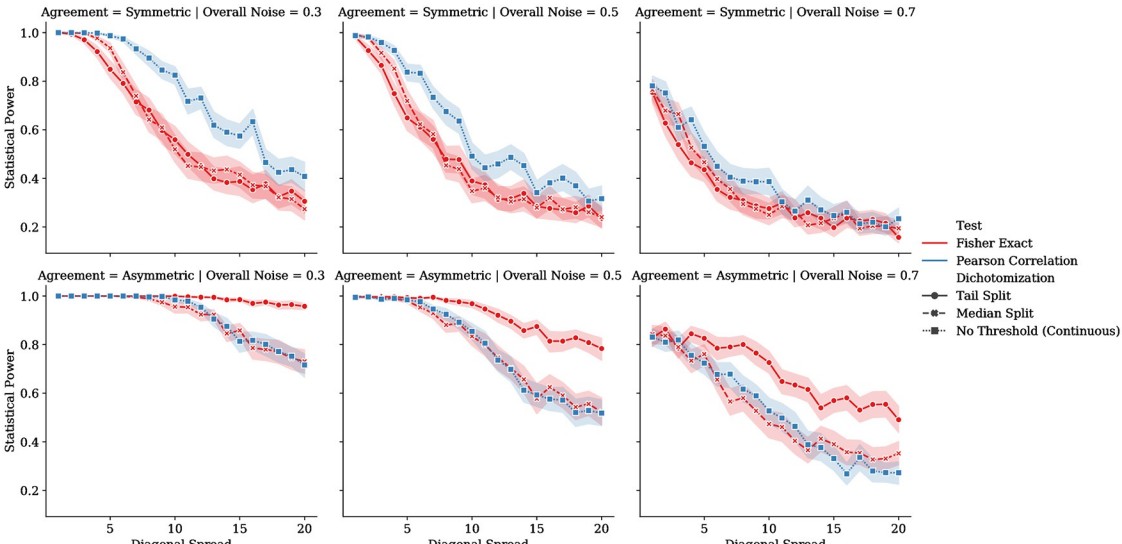

**Fig 5. Statistical power achieved with the Pearson coefficient (a continuous measure of association; blue lines) and Fisher's exact test (a measure of association between dichotomized variables; red lines) for synthetic data with symmetrical (upper row) and asymmetrical (lower row) properties with respect to agreement.** Columns correspond to the level of uniform "overall" noise ($\omega$) added to the data, representing prior uncertainty. X-axes represent the diagonal spread ($\sigma$), and the y-axes represent the test's statistical power for the given sample size and estimated effect sizes. Data subjected to Fisher's exact test were dichotomized at either a threshold of 5 (the "Median Split," denoted by '+' markers in red) or 3 (the "Tail Split," denoted by the dot markers in red). For all series, dark lines denote means and the ribbons are 95% confidence intervals over 100 runs.

for dichotomization of continuous variables. These claims are generally stated only briefly, and with less quantitative support than the more numerous mathematical treatments of the problems with dichotomization [9, 10, 16, 17]. However, these more rigorous quantitative analyses typically involve assumptions of symmetrical or Gaussian distributions of the underlying variables in the context of generalized linear modeling (although Irwin & McClelland [10] demonstrated that median splits of asymmetric and bimodal beta distributions is also deleterious). These analyses have led to vigorous generalized denunciation of variable dichotomization across several disciplines, but our current work offers important counterexamples to this narrative [10, 11].

The Alda score is more broadly used as a target variable in both predictive and associative analyses, and not as a predictor variable, which is an important departure from most analyses against dichotomization. Since there is no valid and reliable biomarker of lithium response, these cases must rely on the Alda score-based definition of lithium response as a "ground truth" target variable. In the case of predicting lithium response, where these ground truth labels are collected from multiple raters across different international sites, variation in lithium response scoring patterns across centres might further accentuate the extant between-site heterogeneity.

To this end, inter-individual differences in subjective rating scales may be more informative about the raters than the subjects, and one may wish to use dichotomization to discard this nuisance variance [8, 9, 16]. Doing so means that one turns regression supervised by a dubious target into classification with a more reliable (although coarser) target. Appropriately balancing these considerations may require more thought than adopting a blanket prohibition on dichotomization or some other form of preprocessing.

An important criticism of continuous variable dichotomization is that it may impede comparability of results across studies, both in terms of diminishing power and inflating heterogeneity

[17]. However, this is more likely a problem when dichotomization thresholds are established on a study-by-study basis, without considering generalizability from the outset. These arguments do not necessarily apply to the Alda score, since the threshold of 7 has been established across a large consortium with support from both reliability and discrete mixture analysis [1], and is the effective standard split point for this scale [18].

Our study thus provides a unique point of support for the dichotomized Alda score insofar as we show that the retention of MI and frequentist statistical power is likely due to asymmetrical reliability across the range of scores. Our analyses show that there is a range of Alda scores (those identifying good lithium responders; scores $\geq 7$) for which scores correspond more tightly to a consensus-defined gold standard in a large scale international consortium. Conversely, this asymmetry implies that Alda scores at the lower end of the range will carry greater uncertainty (Fig 2). This may be due to the intrinsic structure of the Alda scale, wherein a score of $\leq 3$ may result from 2159 item combinations, while only 79 combinations can yield a score of $\geq 7$. In particular, we showed that tail split dichotomization of the Alda score will be more robust to increases in the prior uncertainty (i.e. the overall level of background "noise" in the relationship between true/observed scores). This feature is important since the sample of raters included in the Alda score's calibration study [1] was relatively small and consisted of individuals involved in ConLiGen centres. It is reasonable to suspect that assessment of Alda score reliability in broader research and clinical settings would add further disagreement-based noise to the inter-rater reliability data. At present, use of the dichotomized scale could confer some robustness to that uncertainty.

More generally our study showed that if reliability of a measure is particularly high at one tail of its range, then a "tail split" dichotomization can outperform even the continuous representation of the variable. This presents an important counterexample to previous authors, such as Cohen [5], Irwin & McClelland [10], and MacCallum et al. [9] who argued that "tail splits" are still worse than median splits. While our study reaffirms these claims in the case of measures whose reliability is constant over the domain (see Fig 4B and the upper row of Fig 5), our analysis of the asymmetrically reliable scenario yields opposite conclusions, favouring a "tail split" dichotomization over both median splits and continuous representations. Tail split dichotomization was particularly robust when data were affected by both asymmetrical reliability *and* high degrees of uniform noise over the variable's range. Together, these results suggest that dichotomization/categorization of a continuous measurement may be justifiable when its relationship to the underlying ground truth variable is noisy everywhere except at an extreme.

Our study has several limitations. First, our sample size for the re-analysis of the Alda score reliability was relatively small, and sourced from highly specialized raters involved in lithium-specific research. However, one may consider this sample as representative of the "best case scenario" for the Alda score's reliability. It is likely that further expansion of the subject population would introduce more noise into the relationship between ground truth and observed Alda scores. It is likely that most of this additional disagreement would be observed for lower Alda scores, since (A) there are simply more potential item combinations that can yield an Alda score of 5 than an Alda score of 9, for example, and (B) unambiguously excellent lithium response is a phenomenon so distinct that some question whether lithium responsive BD may constitute a unique diagnostic entity [19, 20]. Thus, we believe that our sample size for the reliability analysis is likely sufficient to yield the present study's conclusions.

Our study is also limited by the fact that theoretical analysis was largely simulation-based, and thus cannot offer the degree of generalizability obtained through rigorous mathematical proof. Nonetheless, our study offers sufficient evidence—in the form of a counterexample—to show that there exist scenarios in which dichotomization is statistically superior to preserving

a variable's continuous representation. Furthermore, we used well controlled experiments to isolate asymmetrical reliability as the cause of dichotomization's superiority across simulated conditions.

## Conclusion

In conclusion, we have shown that a dichotomous representation of the Alda score for lithium responsiveness is more robust to noise arising from inter-rater disagreement. The dichotomous Alda score is therefore likely a better representation of lithium responsiveness for multi-site studies in which lithium response is a target or dependent variable. Through both re-analysis of the Alda score's real-world inter-rater reliability data and careful theoretical simulations, we were able to show that asymmetrical reliability across the score's domain was the likely cause for superiority of the dichotomous definition. Our study is not only important for future research on lithium response, but other studies using subjective and potentially unreliable measures as dependent variables. Practically speaking, our results suggest that it might be better to classify something we can all agree upon than to regress something upon which we can not.

## Supporting information

**S1 Fig. A-score reliability histograms.** Histograms of ratings for each value of the ground truth Alda A-score. This figure was generated identically to Fig 2, but using the A-score data only.
(PDF)

**S2 Fig. A-score mutual information results.** Mutual information between gold standard and observed Alda A-scores in relation to observation noise and the scale's "raw" or dichotomized form. This figure was generated identically to Fig 3, but using the A-score data only.
(PDF)

**S1 File. Total Alda score ratings.** Inter-rater reliability data for the total Alda score.
(CSV)

**S2 File. Alda A-score ratings.** Inter-rater reliability data for the Alda A-score.
(CSV)

**S3 File. Alda score analysis code.** Mathematica notebook containing the empirical evaluation of the Alda Score of Lithium response. This notebook also contains additional analysis of the A-score alone.
(NB)

**S4 File. Theoretical analysis code.** Mathematica notebook containing the theoretical analyses of discrete vs. continuous mutual information in asymmetrically reliable data.
(NB)

**S5 File. Code for statistical power tests.** Jupyter notebook containing the theoretical analyses of the statistical power of classical associative tests under asymmetrically reliable data.
(IPYNB)

**S6 File. Alda score analysis code (PDF version).** PDF version of S3 File for those without Mathematica license.
(PDF)

**S7 File. Theoretical analysis code (PDF version).** PDF version of S4 File for those without Mathematica license.
(PDF)

## Acknowledgments

The authors wish to acknowledge those members of the Consortium on Lithium Genetics (ConLiGen) who contributed ratings for the vignettes herein: Mirko Manchia, Raffaella Ardau, Jean-Michel Aubry, Lena Backlund, Claudio E.M. Banzato, Bernhard T. Baune, Frank Bellivier, Susanne Bengesser, Clara Brichant-Petitjean, Elise Bui, Cynthia V. Calkin, Andrew Tai Ann Cheng, Caterina Chillotti, Scott Clark, Piotr M. Czerski, Clarissa Dantas, Maria Del Zompo, J. Raymond DePaulo, Bruno Etain, Peter Falkai, Louise Frisén, Mark A. Frye, Jan Fullerton, Sébastien Gard, Julie Garnham, Fernando S. Goes, Paul Grof, Oliver Gruber, Ryota Hashimoto, Joanna Hauser, Rebecca Hoban, Stéphane Jamain, Jean-Pierre Kahn, Layla Kassem, Tadafumi Kato, John R. Kelsoe, Sarah Kittel-Schneider, Sebastian Kliwicki, Po-Hsiu Kuo, Ichiro Kusumi, Gonzalo Laje, Catharina Lavebratt, Marion Leboyer, Susan G. Leckband, Carlos A. López Jaramillo, Mario Maj, Alain Malafosse, Lina Martinsson, Takuya Masui, Philip B. Mitchell, Frank Mondimore, Palmiero Monteleone, Audrey Nallet, Maria Neuner, Tomás Novák, Claire O'Donovan, Urban Ösby, Norio Ozaki, Roy H. Perlis, Andrea Pfennig, James B. Potash, Daniela Reich-Erkelenz, Andreas Reif, Eva Reininghaus, Sara Richardson, Janusz K. Rybakowski31, Martin Schalling, Peter R. Schofield, Oliver K. Schubert, Barbara Schweizer, Florian Seemüller, Maria Grigoroiu-Serbanescu, Giovanni Severino, Lisa R. Seymour, Claire Slaney, Jordan W. Smoller, Alessio Squassina, Thomas Stamm, Pavla Stopkova, Sarah K. Tighe, Alfonso Tortorella, Adam Wright, David Zilles, Michael Bauer, Marcella Rietschel, and Thomas G. Schulze.

## Author Contributions

**Conceptualization:** Abraham Nunes.

**Data curation:** Martin Alda.

**Formal analysis:** Abraham Nunes.

**Investigation:** Abraham Nunes.

**Methodology:** Abraham Nunes, Martin Alda.

**Resources:** Martin Alda.

**Software:** Abraham Nunes.

**Supervision:** Thomas Trappenberg, Martin Alda.

**Validation:** Abraham Nunes.

**Visualization:** Abraham Nunes.

**Writing – original draft:** Abraham Nunes.

**Writing – review & editing:** Abraham Nunes, Thomas Trappenberg, Martin Alda.

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
