## [Decision Letter · Decision Letter 0]

10 Dec 2019

PONE-D-19-30408

Asymmetrical reliability of the Alda Score favours a dichotomous representation of lithium responsiveness

PLOS ONE

Dear Dr. Alda,

Thank you for submitting your manuscript to PLOS ONE. After careful consideration, we feel that it has merit but does not fully meet PLOS ONE’s publication criteria as it currently stands. Therefore, we invite you to submit a revised version of the manuscript that addresses the points raised during the review process.

We would appreciate receiving your revised manuscript by Jan 24 2020 11:59PM. To enhance the reproducibility of your results, we recommend that if applicable you deposit your laboratory protocols in protocols.io, where a protocol can be assigned its own identifier (DOI) such that it can be cited independently in the future. For instructions see: http://journals.plos.org/plosone/s/submission-guidelines#loc-laboratory-protocols

We look forward to receiving your revised manuscript.

Kind regards,

Vincenzo De Luca

Academic Editor

PLOS ONE

Journal Requirements:

Reviewers' comments:

Reviewer's Responses to Questions

**Comments to the Author**

1. Is the manuscript technically sound, and do the data support the conclusions?

Reviewer #1: Yes

Reviewer #2: Yes

Reviewer #3: Yes

2. Has the statistical analysis been performed appropriately and rigorously? 

Reviewer #1: Yes

Reviewer #2: Yes

Reviewer #3: Yes

3. Have the authors made all data underlying the findings in their manuscript fully available?

Reviewer #1: Yes

Reviewer #2: Yes

Reviewer #3: Yes

4. Is the manuscript presented in an intelligible fashion and written in standard English?

Reviewer #1: Yes

Reviewer #2: Yes

Reviewer #3: Yes

5. Review Comments to the Author

Reviewer #1: This is a very impactful study in the field of lithium research, as it provides robust and supported evidence toward the use of the dichotomization of the Alda scale in a research setting.

The referee has no major comments. One minor suggestion, which is more a curiosity, would be to shortly comment on the performance of the scale at the very low end (i.e. patients with a score of 0 or 1). While this can be inferred from the analysis and tables, readers might benefit from such a comment.

Reviewer #2: This is an interesting paper decribing how the asymmetrical reliability of the Alda Score favours a dichotomous representation of lithium responsiveness. The paper is well-written and the argument is exhaustively presented.

Reviewer #3: This is a relevant and methodologically sound paper exploring the hypothesis that agreement on the assessment of lithium response with a validated and well established tool (the Alda scale)

is higher at the upper end of the score range, and that this asymmetric

agreement is a scenario in which dichotomization of the score is more informative than the

raw measure. To this end the authors perform an empirical re-analysis of a previously published paper as well as analyses of simulated data with varying levels of asymmetrical inter-rater reliability.

I have only minor comments:

- The concept of mutual information should be explained

- Why a Multinomial-Dirichlet model?

6. PLOS authors have the option to publish the peer review history of their article (what does this mean?). If published, this will include your full peer review and any attached files.

Reviewer #1: No

Reviewer #2: Yes: Luca Steardo jr

Reviewer #3: No

---

## [Author Response · Author response to Decision Letter 0]

23 Dec 2019

We thank the reviewers for their consideration and thoughtful critique, which we address below

in a point-by-point fashion. Where comments overlap, we refer to the earliest response.

Reviewer 1:

Concern 1

One minor suggestion, which is more a curiosity, would be to shortly comment on the performance of the scale at

the very low end (i.e. patients with a score of 0 or 1). While this can be inferred from the analysis and tables,

readers might benefit from such a comment.

Thank you. We have added the following statement to the discussion.

“Conversely, this asymmetry implies that Alda scores at the lower end of the range will carry greater uncertainty

(Fig 2). This may be due to the intrinsic structure of the Alda scale, wherein a score of ≤ 3 may result from 2159

item combinations, while only 79 combinations can yield a score of ≥ 7 .”

Reviewer 3:

Concern 1

The concept of mutual information should be explained.

Thank you. We have added the following clarification in the methods section.

“Mutual information is a general measure of dependence that expresses the degree to which uncertainty about

one variable is reduced by observation of another. Whereas the correlation coefficient depends on the existence

of a linear association, MI can detect nonlinear relationships between variables by comparing their joint

probability against the product of their marginal distributions.”

---

## [Decision Letter · Decision Letter 1]

30 Dec 2019

Asymmetrical reliability of the Alda Score favours a dichotomous representation of lithium responsiveness

PONE-D-19-30408R1

Dear Dr. Alda,

We are pleased to inform you that your manuscript has been judged scientifically suitable for publication and will be formally accepted for publication once it complies with all outstanding technical requirements.

With kind regards,

Vincenzo De Luca

Academic Editor

PLOS ONE

Additional Editor Comments (optional):

Reviewers' comments:

Reviewer's Responses to Questions

**Comments to the Author**

1. If the authors have adequately addressed your comments raised in a previous round of review and you feel that this manuscript is now acceptable for publication, you may indicate that here to bypass the “Comments to the Author” section, enter your conflict of interest statement in the “Confidential to Editor” section, and submit your "Accept" recommendation.

Reviewer #1: All comments have been addressed

Reviewer #3: All comments have been addressed

2. Is the manuscript technically sound, and do the data support the conclusions?

Reviewer #1: Yes

Reviewer #3: Yes

3. Has the statistical analysis been performed appropriately and rigorously? 

Reviewer #1: Yes

Reviewer #3: Yes

4. Have the authors made all data underlying the findings in their manuscript fully available?

Reviewer #1: Yes

Reviewer #3: Yes

5. Is the manuscript presented in an intelligible fashion and written in standard English?

Reviewer #1: Yes

Reviewer #3: Yes

6. Review Comments to the Author

Reviewer #1: The authors addressed all the issues raised by the referee. The manuscript can be published in its current form.

Reviewer #3: The authors have addressed all the reviewer's concerns significantly improving the manuscript. I have no further comments.

7. PLOS authors have the option to publish the peer review history of their article (what does this mean?). If published, this will include your full peer review and any attached files.

Reviewer #1: No

Reviewer #3: No

---

## [Editor Report · Acceptance letter]

2 Jan 2020

PONE-D-19-30408R1 

Asymmetrical reliability of the Alda Score favours a dichotomous representation of lithium responsiveness 

Dear Dr. Alda:

I am pleased to inform you that your manuscript has been deemed suitable for publication in PLOS ONE. Congratulations! Your manuscript is now with our production department. 

With kind regards,

on behalf of

Dr. Vincenzo De Luca 

Academic Editor

PLOS ONE